# Long COVID and its associated factors among COVID survivors in the community from a middle-income country—An online cross-sectional study

**Foong Ming Moy**[1]\*, **Noran Naqiah Hairi**[1], **Eugene Ri Jian Lim**[2], **Awang Bulgiba**[1]

**1** Department of Social & Preventive Medicine, Centre for Epidemiology & Evidence-Based Practice, Faculty of Medicine, University of Malaya, Kuala Lumpur, Malaysia, **2** School of Medicine, International Medical University, Kuala Lumpur, Malaysia

\* moyfm@ummc.edu.my

**Data Availability Statement:** All relevant data are within the paper and its Supporting Information files.

## Abstract

Patients with COVID-19 usually recover and return to normal health, however some patients may have symptoms that last for weeks or even months after recovery. This persistent state of ill health is known as Long COVID if it continues for more than three months and are not explained by an alternative diagnosis. Long Covid has been overlooked, especially in the low- and middle-income countries. Therefore, we conducted an online survey among the COVID-19 survivors in the community to explore their Long COVID symptoms, factors associated with Long COVID and how Long COVID affected their work. A total of 732 COVID-19 survivors responded, with 56% were without or with mild symptoms during their acute COVID-19 conditions. One in five COVID-19 survivors reported of experiencing Long COVID. The most commonly reported symptoms were fatigue, brain fog, depression, anxiety and insomnia. Females had 58% higher odds (95% CI: 1.02, 2.45) of experiencing Long COVID. Patients with moderate and severe levels of acute COVID-19 symptoms had OR of 3.01 (95% CI: 1.21, 7.47) and 3.62 (95% CI: 1.31, 10.03) respectively for Long COVID. Recognition of Long COVID and its associated factors is important in planning prevention, rehabilitation, clinical management to improve recovery from COVID-19.

## Introduction

As of July 2022, Malaysia, a middle-income country has had more than 4.5 million individuals infected with COVID-19 with close to 36 thousand deaths since March 2020. While most patients with COVID-19 recover and return to normal health, some patients may have symptoms that last for weeks or even months after recovery from COVID-19. This persistent state of ill health is known as Long COVID or post COVID condition if it continues for more than 12 weeks and are not explained by an alternative diagnosis [1].

Patients with Long COVID reported experiencing different combinations of symptoms such as fatigue, dyspnoea, cough, anxiety, cognitive impairment, myalgia, symptoms that get

**Funding:** This study is part of the COVID-19 Epidemiological Analysis and Strategies (CEASe) Project with funding from the Ministry of Science, Technology and Innovation (UM.0000245/HGA. GV). The funders had no role in study design, data collection and analysis, decision to publish, or preparation of the manuscript.

**Competing interests:** The authors have declared that no competing interests exist.

worse after physical activities [2,3]. These symptoms could be driven by a direct effect of virus infection and might be explained by several hypotheses including aberrant immune response, hyperactivation of the immune system, or autoimmunity [4]. Additionally, indirect effects including reduced social contact, loneliness, incomplete recovery of physical health, and loss of employment could affect psychiatric symptoms [5].

Preliminary results from a national survey in the UK estimated that around one in ten COVID-19 patients exhibited symptoms for a period of 12 weeks or longer [6]. Another study found that 30% of COVID-19 patients surveyed still had persistent symptoms after nine months [7]. This has become an increasing concern. Most patients had good physical and functional recovery during follow-up, and the majority of them who were employed before COVID-19 had returned to their original work. However, some may still experience poorer work performance due to the Long COVID sequalae [4,5].

Studies on Long Covid were mainly conducted in Europe and on patients recently discharged from hospitals. A living systematic review identified limited evidence from low to middle-income countries and among people who were not hospitalized [3]. In Malaysia, Long Covid has been overlooked and less emphasis was given to the issue. There is a scarcity of findings on Long COVID. To date, there are only some preliminary findings among COVID-19 survivors, reported by the Ministry of Health Malaysia. Based on the Sungai Buloh COVID-19 Rehabilitation Outpatient Specialized Services databases, after 12 weeks of acute COVID-19, 474 (63.6%) out of 745 COVID-19 survivors experienced post COVID-19 syndrome. The five most commonly reported symptoms were fatigue (73.4%); exertional dyspnoea (19.4%); insomnia (13.9%); cough (9.7%) and pain (7.3%). However, psychological symptoms such as anxiety, depression and stress were not reported. In addition, information on whether COVID-19 survivors who experienced milder symptoms or those not admitted to hospitals experienced any Long COVID symptoms and what are the factors associated with Long COVID are not available. Therefore, we conducted an online survey among the COVID-19 survivors in the community to explore their Long COVID symptoms, factors associated with Long COVID and how Long COVID affected their work.

## Methods

This was a cross sectional study conducted from July to September 2021, during the implementation of a nationwide movement control order (MCO) or lockdown. Data was collected using an online questionnaire, the REDCap electronic data capture tools hosted at our university (University of Malaya). REDCap (Research Electronic Data Capture) [8,9] is a secure, web-based software platform designed to support data capture for research studies, providing a) an intuitive interface for validated data capture; b) audit trails for tracking data manipulation and export procedures; c) automated export procedures for seamless data downloads to common statistical packages; and d) procedures for data integration and interoperability with external sources.

The questionnaire was prepared in English, and translated into the Malay and Chinese languages to cater for the multiethnic population. It was distributed in social media, COVID-19 support group webpage, news media in the above-mentioned languages with the aim of reaching as many COVID-19 patients as possible. The questionnaire was anonymous and identifiable information such as names, telephone numbers and emails were not collected. However, respondents were given a choice if they wanted to be contacted (to leave their emails) if the researchers found their responses to be in need of referral for further attention, or for follow up with additional activities or assistance required.

Ethics clearance was obtained from the University Malaya Research Ethics Committee (Reference number: UM.TNC2/UMREC_1439). Written informed consent was obtained before data collection. The questionnaire covers information such as socio-demographic characteristics, existing comorbidities, self-perception on health, information on the acute COVID-19 condition and treatment received, symptoms and duration of post-COVID condition, effects on occupation and mental health status.

Data from RedCap were exported into the SPSS version 23 software for data analysis. Mean with standard deviation (SD) or median with inter-quartile range (IQR) was used to report normally distributed or skewed data respectively. Frequency with percentage was used to present categorical variables. Univariable and multivariable logistic regression analyses were carried out to determine the factors associated with Long COVID. Variables with $p$ values < 0.25 or clinically important variables were included in the final model. Significance level was preset at $p < 0.05$.

## Results

A total of 732 COVID survivors responded in the online survey. There were slightly more females (58.7%) who responded (Table 1). More than 60% of the respondents were in the twenties and thirties with tertiary education levels (Degree and above). Two third of the respondents resided in the central region of Peninsular Malaysia. More than half of them were overweight. About 74% of them were free of comorbidities. Majority of them were working. Only one quarter of them had full vaccination at the time of survey.

Among these respondents, about 56% were without or with mild symptoms during their acute COVID-19 conditions (Table 2). Almost half of them were on home quarantine, with the others admitted to the COVID-19 centres or hospitals. Among those hospitalised, 20% required the use of ventilators while another 23% were admitted to Intensive Care Units (ICU). The mean duration of hospital stay was 10.9±8.2 days with a minimum of 2 days and maximum of 52 days. The mean duration since the diagnosis of COVID-19 was 27.3 ± 12.5 weeks, with more than 95% were diagnosed at least 3 months ago.

Post-COVID symptoms were experienced at different durations. Cumulatively, approximately 88% experienced post-COVID symptoms up to 6 weeks since diagnosis, 48% more than 6 weeks, 21.1% more than 3 months (Long COVID) and 10% more than 6 months. Fig 1 presents the common symptoms reported over the duration mentioned. The most commonly reported symptoms for Long COVID (more than 3 months) were fatigue, brain fog, depression, anxiety, insomnia, arthralgia or myalgia. These commonly reported symptoms are similar at duration up to 6 weeks and more than 6 weeks.

Among those working, one third of them reported that their work performance was affected with majority of them (73%) reduced their working hours, while some (23.9%) had to take leave or even resign from their jobs (3%). Only 67.5% of them perceived to be in good health currently compared to before they were infected with COVID-19 (92.9%) (Table 3).

At univariate level, only severity of acute COVID-19 was significantly associated with Long COVID; while sex and comorbidities were marginally associated; age groups and BMI categories were not associated with Long COVID (Table 3). In the multivariable models, females were found to have 58% higher odds of experiencing Long COVID. Patients with moderate and severe levels of acute COVID-19 had OR of 3.01 (95% CI: 1.21, 7.47) and 3.62 (95% CI: 1.31, 10.03) respectively for Long COVID (Table 4).

## Discussion

As we used an online questionnaire to collect data, the respondents were younger and with higher education levels. Almost two-thirds were from the central region of Peninsular

**Table 1. Socio-demographic characteristics, Body Mass Index (BMI), vaccination status and medical history of respondents.**

| n = 732 | | n (%) |
|---|---|---|
| Sex | Males | 302 (41.3) |
| | Females | 430 (58.7) |
| Age groups | twenties | 156 (25.5) |
| (n = 611) | thirties | 224 (36.7) |
| | forties | 33 (5.4) |
| Mean age = 40.2±10.9 | fifties | 84 (13.7) |
| | sixties | 114 (18.7) |
| Education levels | Secondary and lower | 103 (14.1) |
| | Diploma | 142 (19.4) |
| | Bachelor's degree | 341 (46.6) |
| | Master's degree or higher | 146 (19.9) |
| Regions of residence | | |
| (n = 729) | Northern | 52(7.1) |
| | Eastern | 26(3.6) |
| | Central | 543(74.5) |
| | Southern | 71(9.7) |
| | East Malaysia | 37(5.1) |
| Currently working | Yes | 618 (84.4) |
| Vaccination | No | 341 (46.6) |
| | One dose | 202 (27.6) |
| | Two doses | 189 (25.8) |
| Smoking currently | Yes | 50 (6.8) |
| BMI* categories | | |
| (n = 727) | Underweight | 40 (5.5) |
| | Normal weight | 281 (38.7) |
| | Overweight | 219 (30.1) |
| | Obese | 187 (25.7) |
| Comorbidities** | Free of comorbidities | 541 (73.9) |
| (n = 728) | Diabetes mellitus | 50 (6.9) |
| | Hypertension | 109 (15.0) |
| | Heart disease | 16 (2.2) |
| | Hypercholesterolemia | 123 (16.9) |
| | Cancers | 10 (1.4) |

*underweight ($<18.5kg/m^2$), normal weight ($18.5–24.9kg/m^2$), overweight ($25.0–29.9kg/m^2$), obese ($\geq30.0kg/m^2$).

**the numbers did not add up to 100% as some respondents experienced more than one comorbidity.

Malaysia, the most progressive states, with high population density and where the capital city is situated. Older individuals and those with lower education levels were under-represented. There were also slightly more females who responded to the study, similarly found from other studies with more female participation [10].

The prevalence of comorbidities such as hypertension, diabetes mellitus among the respondents was low as more than half of them were in their thirties and forties. About two third of them were free of comorbidities. Majority of them worked as only 18% of them were in their sixties. However, their BMI status was comparable with the national statistics [11] where half of them were overweight or obese. As our survey was conducted before vaccination was widely

**Table 2. Acute COVID-19 information of respondents.**

| | | n (%) |
|---|---|---|
| Severity of acute COVID-19 n = 634 | No symptom | 71(11.2) |
| | Mild | 287 (45.3) |
| | Moderate | 202 (31.9) |
| | Severe (with oxygen) | 74 (11.7) |
| Care received for acute COVID-19 (n = 638) | Home quarantine | 308 (48.3) |
| | COVID-19 centre | 165 (25.9) |
| | Hospital | 165 (25.9) |
| For those hospitalized (n = 165) | ICU | 33 (20.0) |
| | Ventilator usage | 38 (23.0) |
| Duration since COVID-19 diagnosis (n = 636) | < = 6 weeks | 11(1.7) |
| | >6 to 12 weeks | 16 (2.5) |
| | >3 to 6 months | 269(42.3) |
| | > 6 months | 340(53.5) |
| Post-COVID symptoms (cumulative) (n = 598) | Up to 6 weeks | 528 (88.3) |
| | > 6 weeks | 287 (48.0) |
| | > 3 months (Long COVID) | 126 (21.1) |
| | > 6 months | 60 (10.0) |

available, only one quarter of them reported to be fully vaccinated at the time of survey. However, we were unable to ascertain if the respondents were vaccinated before or after the diagnosis of COVID-19. Hence, the effect of vaccination on Long COVID symptoms or duration was not studied.

According to the COVID-19 Management Guidelines [12], the confirmed COVID-19 patients in Malaysia are classified and explicitly managed based on five categories: stage 1 (asymptomatic), stage 2 (symptomatic without pneumonia), stage 3 (symptomatic with pneumonia), stage 4 (symptomatic with pneumonia and require supplemental oxygen), and stage 5 (critically ill with multi-organ involvement). Based on the meta-analysis by Ng et al [13] on the prevalence of the clinical stages of COVID-19 in Malaysia: 27.8% of the patients were asymptomatic (stage 1), 32% mild (stage 2), 17% moderate (stage 3) and 11% severe (stage 4 and 5). However, only 11% of our respondents were asymptomatic and about one third of our respondents were in the moderate categories. This observed difference may be due to volunteer bias where those in more severe COVID condition were more likely to respond as they may want to report their experiences. We did not report our findings using the 5-stage categories as high proportion of our respondents could not recall this information.

Twenty one percent of our respondents reported to have experienced Long COVID. In another cohort study among hospitalized patients, 76% of patients reported at least one symptom at 6 months after acute COVID-19 [14]. According to a retrospective cohort study using linked electronic health records (EHRs) data of 273,618 COVID-19 survivors, one in three patients had one or more symptoms of long-COVID recorded between 3 and 6 months after a diagnosis of COVID-19 [15]. In another systematic review among patients with mild COVID-19 infection [16], they were reported to experience persistent symptoms after three weeks with the frequency between 10% to 35%. The difference in this reported prevalence maybe

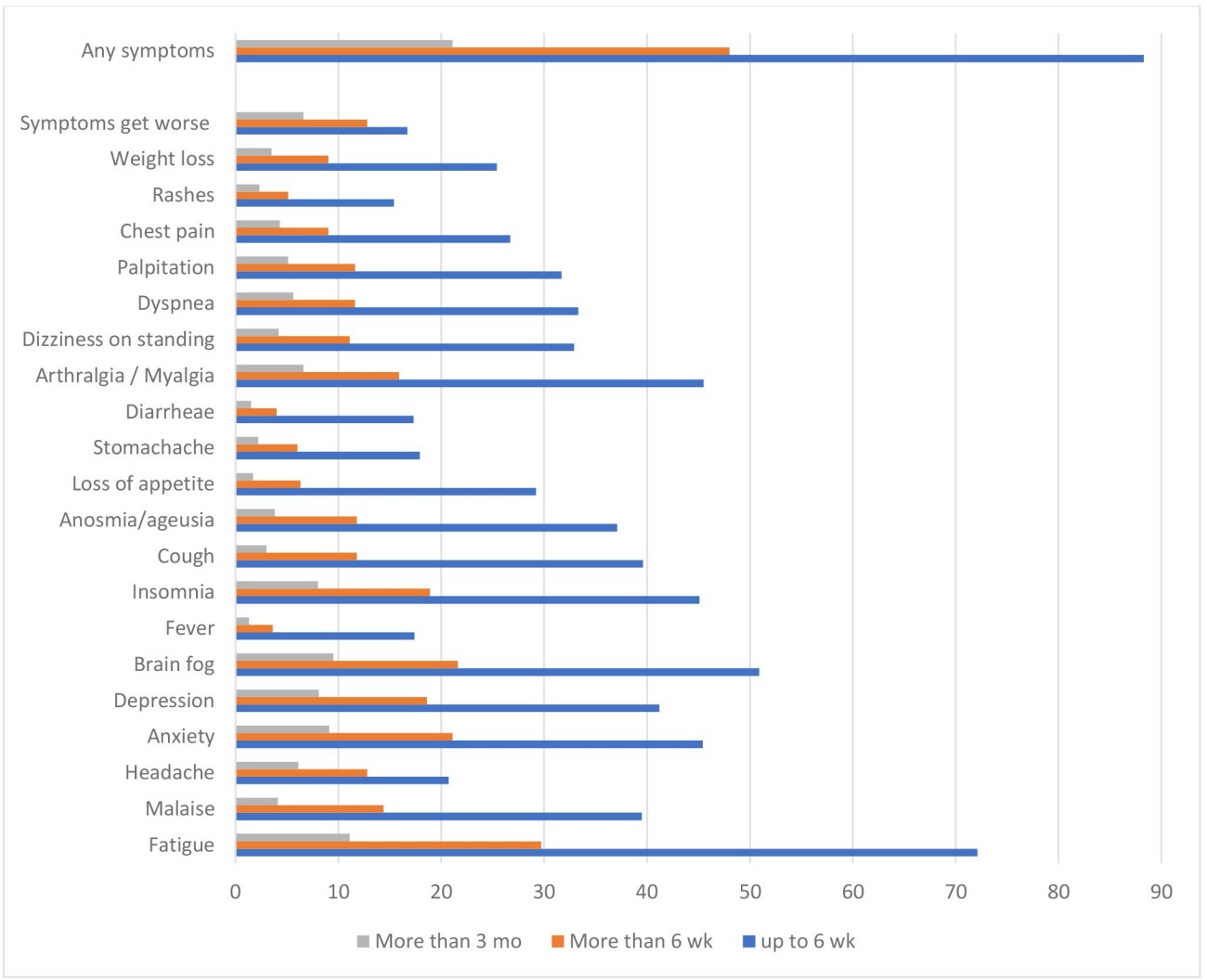

**Fig 1. Post COVID symptoms at different duration.**

influenced by the variety of definitions of Long COVID, study settings and characteristics of patients.

The most commonly experienced symptoms were fatigue, brain fog, depression, anxiety, insomnia, arthralgia or myalgia, which are similarly reported in systematic reviews on Long

**Table 3. Perceived health and work productivity among Long COVID respondents.**

|  |  | n (%) |
|---|---|---|
| Perceived to be in good health | Current | 494 (67.5) |
| n = 732 | Pre-COVID | 680 (92.9) |
| Affected work performance | Yes | 194 (35.3) |
| n = 550 | No | 356 (64.7) |
| Measures taken | Quit work | 6 (3.0) |
| n = 194 | Reduced work hours | 142 (73.1) |
|  | Took leave from work | 46(23.9) |

**Table 4. Factors associated with Long COVID.**

| N = 598 | Long COVID | | p-value | Crude OR (95% CI) | *Adjusted OR (95%CI) |
|---|---|---|---|---|---|
| | Yes (n = 126) | No (n = 472) | | | |
| **Sex** | | | | | |
| Male | 42 (17.4) | 199 (82.6) | 0.073 | Reference | Reference |
| Female | 84 (23.5) | 273 (76.5) | | 1.46 (0.96, 2.20) | 1.58 (1.02, 2.45) |
| **Age group** | | | | | |
| Twenties | 28 (22.2) | 98 (77.8) | 0.492 | Reference | |
| Thirties | 44 (23.3) | 145 (76.7) | | 1.06 (0.62, 1.82) | |
| Forties | 5 (20.0) | 20 (80.0) | | 0.87 (0.30, 2.54) | |
| Fifties | 12 (20.3) | 47 (79.7) | | 0.89 (0.42, 1.91) | |
| Sixties | 14 (14.3) | 84 (85.7) | | 0.58 (0.29, 1.18) | |
| **BMI categories** | | | | | |
| Under-Normal weight | 51 (19.5) | 210 (80.5) | 0.413 | Reference | Reference |
| Overweight | 37 (20.7) | 142 (79.3) | | 1.07 (0.67, 1.72) | 1.12 (0.68, 1.83) |
| Obese | 38 (25.0) | 114 (75.0) | | 1.37 (0.85, 2.21) | 1.14 (0.69, 1.90) |
| **Any comorbidities** | | | | | |
| Yes | 43 (26.4) | 120 (73.6) | 0.051 | 1.52 (0.99, 2.32) | 1.43 (0.91, 2.25) |
| No | 83 (19.1) | 352 (80.9) | | Reference | Reference |
| **Severity of acute COVID** | | | | | |
| No symptom | 6 (10) | 54 (90.0) | 0.003 | Reference | Reference |
| Mild | 48 (17.5) | 227 (82.5) | | 1.90 (0.77, 4.68) | 1.82 (0.74, 4.49) |
| Moderate | 51 (26.7) | 140 (73.3) | | 3.28 (1.33, 8.08) | 3.01 (1.21, 7.47) |
| Severe (with oxygen) | 21 (30.4) | 48 (69.6) | | 3.94 (1.47, 10.57) | 3.62 (1.31, 10.03) |

*Adjusted for sex, age groups, BMI categories, comorbidities and severity of acute COVID.

**Some totals did not add up to 598 due to missing data.

COVID [3,17]. The Long COVID symptoms' aetiology is complex, and several mechanisms might be responsible for these variety of symptoms. It appears that persistent inflammation plays a key role in the pathogenesis of Long COVID symptoms. In a study among patients who experienced fatigue, cognitive decline, and apathy after recovering from Covid-19, researchers discovered an increased amount of cytokines in the blood, particularly interleukin-6 (IL-6), which alters neuronal function as it passes through the blood-brain barrier, resulting in neurological complications of Long COVID [18].

Covid-19 is also shown to be neurotropic, which contributes to a higher occurrence of neurological symptoms as it invades the central nervous system. Damage to the brainstem, which has limited regeneration capacity, might lead to long-term cardiorespiratory and neurological repercussions, which could be the cause of long-term COVID symptoms [19]. Depression is also found to have an association with IL-6, regardless of other factors of depression that occurred during the COVID-19 pandemic.

Other than inflammation, Long COVID fatigue could be caused by lung dysfunction. This owes to the acute viral infection that initiated the inflammatory response, which subsequently caused fibrosis to the lungs. The severity of the fibrosis would manifest a spectrum of clinical manifestations such as fatigue, cognitive symptoms and breathlessness [20]. As the gut plays a role in controlling the body's immune system, gut microbial dysfunction has also been linked to gastrointestinal and neurological symptoms of Long COVID [21].

Anxiety, depression and stress were reported by a great proportion of our respondents. This could be due to their worries about their delayed recovery. The aetiology of the psychiatric consequences of COVID-19 infection is likely to be multifactorial and might include the direct effects of viral infection, cerebrovascular disease, the degree of physiological compromise, the immunological response, medical interventions, social isolation, the psychological impact of the severe and potentially fatal COVID-19, concerns about infecting others, and stigma [5]. The link between inflammation with increased concentrations of C-reactive protein, ferritin, and interleukin-6; with depression might also explain some of the psychiatric morbidity [22].

Due to the post-COVID symptoms, respondents who were working reported that their work performance or productivity was affected, where they had to reduce their working hours or take leave. Davis et al [23] reported that 45.2% of their Long COVID participants required a reduced work schedule compared to pre-illness, and an additional 22.3% were not working at the time of survey due to illness. Aiyegbusi et al also reported that those who were working before getting infected were unable to return to work due to chronic symptoms, while those who were able to return to work had to modify their responsibilities or cut their working hours due to health concerns, and there are also those who were previously hospitalized that were unable to return to work [24]. Symptoms like brain fog or difficult to concentrate may affect the work performance of those working in the education or accounting sectors. These affected individuals may take longer time to return to their original level of performance. Being lethargic may reduce their work performance. Employers and their fellow colleagues should be more empathetic with their situation and provide support when required.

Based on previous literature, sex, BMI categories, comorbidities and severity of acute COVID-19 were reported to be associated with Long COVID [3,19,25]. However, we only found sex and severity of acute COVID-19 to be significantly associated with Long COVID. Females had 58% higher odds of getting Long COVID, as reported elsewhere [15,26]. According to the autoimmune hypothesis, females have stronger immune response than males due to genetic and hormonal factors. This contributes to a more active immune response where activation of white blood cells, production of inflammatory markers and antibodies are stronger than males. This could be seen as a double-edged sword as it appears protective towards severe symptoms and deaths from COVID-19, but it could bring about the emergence of autoimmune inflammatory symptoms in Long COVID [27].

It should be noted that 10 to 17% of our respondents without symptoms or with mild symptoms experienced Long COVID. The available evidence on the occurrence of post-COVID symptoms in individuals with mild Covid-19 infection is minimal. The majority of patients with mild COVID infections are usually treated as outpatients at home or in clinics, with minimal care devoted in comparison to those who are admitted to the hospital with serious infections [16]. Therefore, the post-COVID or Long COVID symptoms may not be captured by the health authorities if the patients did not seek treatment. These patients should be educated on Long COVID symptoms and to seek treatment if needed. The Ministry of Health Malaysia has a protocol on the management of Long COVID for the primary health care facilities [28]. With that, it is hope that the primary care physicians will take appropriate action among COVID-19 survivors who reported post-COVID symptoms.

On the other hand, respondents who had moderate to severe acute COVID-19 condition had higher (3.0 to 3.6 times) odds of experiencing Long COVID, compared to those without symptoms. Similar results are reported by Taquet at al [15]. This relationship may be explained by the immune response to the SARS-CoV-2 virus which stimulates the production of cytokines and other inflammatory mediators, with higher concentrations found in those with a

more severe COVID-19 [26]. The multi-systemic inflammatory response to the virus may also be responsible for persistent COVID-19 symptoms in survivors [29,30].

The prevalence of Long COVID increased with age, almost doubling the prevalence of Long COVID for those more than 70 years old compared to those 18 to 49 years old [31]. As an individual ages, one's ability to manage viral load also deteriorates with age. The inflammatory process may be maladaptive, causing immune system hyperactivation and small blood vessel hypercoagulation, contributing to multi-system involvement in Long COVID [32]. However, age group was not found to be associated with Long COVID in our study as most of our respondents were young, probably the senior citizens may not be well versed in technology and could not take part in our study since we used a self-reported online questionnaire.

Being overweight or obese predisposed to post-COVID symptoms. Obese COVID survivors usually take longer time to resolve abnormalities on chest radiographs, which is linked to symptoms of Long COVID, such as fatigue, cognitive decline and shortness of breath etc [33–35]. These obese COVID survivors may have clustering of risk factors such as old age with multiple comorbidities. However, we did not find an association between BMI categories with Long COVID, probably our sample were young and with lower rates of comorbidities.

Some studies showed no correlation between the prolonged COVID symptoms and comorbidities of patients [25,33], similar with our findings. However, comorbidities such as hypertension, diabetes and dyslipidemia could be indirect risk factors to Long COVID as they are commonly associated with obesity that might increase the likelihood of Long COVID [25,34].

While interpreting the results, there are some limitations that need to be addressed. First, this is a cross sectional study where respondents were asked to recall their symptoms and duration of experience. This may subject to recall bias and it is impossible to retrospectively confirm the length of COVID-19 related symptoms. In order to overcome these issues, a prospective cohort study may be a better study design. We also did not assess whether the symptoms reported were intermittent or continuous for the entire three or six months. Next, selection bias cannot be avoided in an online study as younger, more highly educated and tech-savvy individuals were more likely to respond. Therefore, our respondents may not be fully representative of the general population. We could not ascertain the effect of vaccination on Long COVID. Some studies suggested that vaccination may lead to potentially lower prevalence of Long Covid [36]. These should be explored in future studies.

The definition of Long COVID used in the current study may be different from the latest definition proposed by WHO on the 6 October 2021 [37]. The latest definition defines Long COVID or Post COVID-19 condition as condition that occurs in individuals with a history of probable or confirmed SARS-CoV-2 infection, usually 3 months from the onset of COVID-19 with symptoms that last for at least 2 months and cannot be explained by an alternative diagnosis. The difference between the two definitions is that the symptoms should last for at least 2 months, which is missing from the earlier definition. However, this cannot be rectified as our study was completed before the new definition was announced.

The pathophysiology and mechanism behind the persistence of symptoms or Long Covid has yet to be identified as researchers are still trying to learn and understand about COVID-19 which is a new disease and Long COVID, an even newer condition. To our knowledge, this represents the first study to describe the symptoms, duration and associated factors of Long COVID in Malaysia. With over four million COVID 19 positive cases in Malaysia at the time of writing, and with numbers likely to increase due to the Omicron variant, the number of survivors with Long COVID will continue to increase. This could have significant and lasting health and economic consequences for the country. Our findings can help in improve planning for the use of relevant healthcare resources.

## Conclusion

This study provides additional insight on the symptoms and duration of post-COVID symptoms as well as the associated factors with Long COVID among COVID-19 survivors in Malaysia. Recognition of Long COVID and its associated factors is important in planning prevention, rehabilitation, clinical management to improve recovery and long-term COVID-19 outcomes. With recurrent and repeated COVID-19 infections, a more comprehensive investigation is necessary to understand the impact of Long COVID.

## Supporting information

**S1 Data.**
(XLSX)

## Acknowledgments

The authors would like to thank the respondents in sharing their experience in the study. The support from the University Malaya is acknowledged.

## Author Contributions

**Conceptualization:** Foong Ming Moy, Noran Naqiah Hairi, Awang Bulgiba.

**Data curation:** Foong Ming Moy.

**Formal analysis:** Foong Ming Moy.

**Funding acquisition:** Awang Bulgiba.

**Investigation:** Foong Ming Moy.

**Methodology:** Foong Ming Moy, Noran Naqiah Hairi, Awang Bulgiba.

**Project administration:** Foong Ming Moy.

**Resources:** Foong Ming Moy, Noran Naqiah Hairi.

**Writing – original draft:** Foong Ming Moy, Eugene Ri Jian Lim.

**Writing – review & editing:** Foong Ming Moy, Noran Naqiah Hairi, Eugene Ri Jian Lim, Awang Bulgiba.

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
