## [Decision Letter · Decision Letter 0]

13 Jul 2022

PONE-D-22-13688Long COVID and its associated factors among COVID survivors in the community from a middle-income country: an online cross-sectional studyPLOS ONE

Dear Dr. Moy,

Thank you for submitting your manuscript to PLOS ONE. After careful consideration, we feel that it has merit but does not fully meet PLOS ONE’s publication criteria as it currently stands. Therefore, we invite you to submit a revised version of the manuscript that addresses the points raised during the review process.

We look forward to receiving your revised manuscript.

Kind regards,

Sairah Hafeez Kamran, PhD

Academic Editor

PLOS ONE

Journal Requirements:

“This study is part of the COVID-19 Epidemiological Analysis and Strategies (CEASe) Project with funding from the Ministry of Science, Technology and Innovation (UM.0000245/HGA.GV).”

5. Please amend your manuscript to include your abstract after the title page.

Reviewers' comments:

Reviewer's Responses to Questions

**Comments to the Author**

1. Is the manuscript technically sound, and do the data support the conclusions?

Reviewer #1: Yes

Reviewer #2: Yes

2. Has the statistical analysis been performed appropriately and rigorously? 

Reviewer #1: No

Reviewer #2: Yes

3. Have the authors made all data underlying the findings in their manuscript fully available?

Reviewer #1: Yes

Reviewer #2: Yes

4. Is the manuscript presented in an intelligible fashion and written in standard English?

Reviewer #1: Yes

Reviewer #2: Yes

5. Review Comments to the Author

Reviewer #1: Dear EiC,

Thanks for sending me this manuscript.

The survey raises an interesting point despite its local origin.

I have few recommendations to be done before further consideration to Publication:

1) The CRONBACH’S ALPHA should be calculated to test reliability between questions.

2) Why Table 4 is adjusted to sex only ? Why not to age and sex?

3) Also, comparison to same level countries like Egypt in the following paper is highly recommended to be implemented in the discussion section as:

Sabry, Nirmeen et al. “Awareness of the Egyptian public about COVID-19: what we do and do not know.” Informatics for health & social care vol. 46,3 (2021): 244-255. doi:10.1080/17538157.2021.1883029

4) Also, I recommend to add this citation in the discussion section as potential solution for the coming infectious outbreaks: the Pharmacists as Healthcare providers in their Community Pharmacies can act as an important and early screening and testing point for COVID-19 infection such as in this mini-review: https://doi.org/10.1177/08971900211036093

Regards

Reviewer #2: Dear author,

I think the tables can be improved especially table 4.There's repetitive us of the word 'reference; within the table.

Overall, the paper needs some minor editing especially on punctuation. For example Long COVID, Central region, Intensive Care Unit. There's no need for capital letters.

6. PLOS authors have the option to publish the peer review history of their article (what does this mean?). If published, this will include your full peer review and any attached files.

Reviewer #1: No

Reviewer #2: No

---

## [Editor Report · Decision Letter 1]

8 Aug 2022

Long COVID and its associated factors among COVID survivors in the community from a middle-income country: an online cross-sectional study

PONE-D-22-13688R1

Dear Dr. Moy,

We’re pleased to inform you that your manuscript has been judged scientifically suitable for publication and will be formally accepted for publication once it meets all outstanding technical requirements.

Kind regards,

Sairah Hafeez Kamran, PhD

Academic Editor

PLOS ONE

---

## [Editor Report · Acceptance letter]

18 Aug 2022

PONE-D-22-13688R1 

Long COVID and its associated factors among COVID survivors in the community from a middle-income country – an online cross-sectional study 

Dear Dr. Moy:

I'm pleased to inform you that your manuscript has been deemed suitable for publication in PLOS ONE. Congratulations! Your manuscript is now with our production department. 

Kind regards, 

on behalf of

Dr. Sairah Hafeez Kamran 

Academic Editor

PLOS ONE